# Perimenopausal Bone Loss Is Associated with Ovulatory Activity—Results of the PeKnO Study (Perimenopausal Bone Density and Ovulation)

**DOI:** 10.3390/diagnostics12020305

**Published:** 2022-01-25

**Authors:** Teresa Starrach, Anita Santl, Vanadin Regina Seifert-Klauss

**Affiliations:** 1Department of Gynecology, School of Medicine, Technical University of Munich (TUM), Ismaninger Str. 22, 81675 Munich, Germany; Teresa.Starrach@med.uni-muenchen.de (T.S.); post.anita.santl@gmail.com (A.S.); 2Klinik und Poliklinik für Frauenheilkunde und Geburtshilfe, Klinikum Großhadern, Marchioninistr. 15, 81377 Munich, Germany; 3Hausärztliche Gemeinschaftspraxis Peissenberg, Teaching Practice of the TUM, 82380 Peißenberg, Germany

**Keywords:** perimenopause, bone density, bone metabolism, anovulation, ovulation

## Abstract

Background: During the menopausal transition, around 25% of women experience a particularly accelerated loss of bone mineral density. These so-called “fast bone losers” represent a group of patients with an increased risk of osteoporosis. The precise mechanisms underlying this extraordinary level of bone mass reduction have not yet been conclusively elucidated. The PeKnO study (*Perimenopausale Knochendichte und Ovulation*; Perimenopausal Bone Density and Ovulation) was a 2-year prospective study investigating menstrual cycle changes, hormonal levels, markers of bone metabolism, and changes in bone mineral density (BMD) in perimenopausal women. The PeKnO study specifically focused on the questions of when the maximum of bone loss occurs, whether the decreasing number of ovulatory cycles correlates with increased bone density loss, and which hormones play a role during these processes. Methods: Healthy women aged ≥45 years with menstrual cycles of ≤42 days and without any exogenous hormonal intake continually self-assessed the lengths of their menstrual cycles and the occurrence of LH peaks with the help of a commercially available electronic fertility monitoring device. At baseline and at 6, 12, 18, and 24 months, hormones (LH, FSH, 17β-estradiol, progesterone, cortisol) and markers of bone metabolism (bone-specific alkaline phosphatase (BAP), osteocalcin (OC), and CTX (C-terminal telopeptide) were assessed during the luteal phase. Trabecular bone density was measured in the lumbar spine (vertebrae L1 through L3) by means of quantitative computed tomography (QCT) at the beginning and at the end of the 2-year study period. Patients were divided into 3 groups according to the changes in bone mineral density (BMD) that occurred within the period of 2 years: group I with an increase in BMD, group II with a decrease in BMD of ≤7%, and group III with a decline in BMD of >7%. Women in the latter group were defined as fast bone losers. Results: From a total of 72 recruited patients with an average age of 48.1 (±2.4) at baseline, complete 2-year data were available from 49 participants. Over the course of 24 months, mean bone mineral density decreased by −4.26 (±4.65). In the same time period, the proportion of ovulatory cycles declined from 67% to 33%. The decrease in the ovulatory rate significantly correlated with an enhanced BMD loss (r = 0.68; *p* < 0.05). Twelve of the forty-nine participants (24.3%) showed a BMD loss of >7% and were identified as fast bone losers. Levels of the luteal phase hormones LH, FSH, 17β-estradiol, and progesterone were significantly different between the three groups. Conclusion: The PeKnO study confirms a marked decline of the ovulatory rate during perimenopause, which is associated with an increased bone density loss while estrogen levels are still adequate.

## 1. Introduction

### 1.1. Perimenopausal Changes in Bone Mineral Density

Until recently, little was known about the exact onset of menopause-associated bone loss and its early phase, just as little as about the initiating mechanisms involved. Bone mineral density loss around menopause has so far been largely attributed to the occurring estrogen deficiency. The present work confirms that this theory is insufficient as a sole explanation. It reports on observations which may help integrate both major, more recent hypotheses on the initiation of perimenopausal bone loss.

### 1.2. State of the Art of Research on the Causes of Perimenopausal Bone Density Loss

Numerous longitudinal studies have shown an association between declining estradiol levels and bone mineral density loss [1,2,3,4]. As early as in 1996, however, Ebeling et al. were able to demonstrate that increased bone density loss during perimenopause is associated with additional influencing factors other than estradiol, as during this phase, the estradiol supply is still adequate while FSH and LH levels are already rising [5].

In the large SWAN study (Study of Women’s Health Across the Nation), Sowers et al. investigated 2311 pre- and perimenopausal women over a 4-year period and showed that higher baseline and follow-up levels of FSH resulted in lower bone mineral density levels. Only in post-menopause had decreased estradiol concentrations been associated with an accelerated loss of bone mineral density. Sowers et al. postulated that during menopausal transition, FSH levels were more apt to characterize ovarian aging and would thus possibly be better predictors for bone density loss [2,6]. Sun et al., a group of basic scientists in osteology around Samir Zaidi in New York, almost simultaneously postulated that high FSH concentrations might be directly responsible for accelerated bone density loss. His research group was able to demonstrate that FSH receptor-deficient mice show no loss in bone density [7].

Progesterone, on the other hand, which is circulating in more elevated concentrations only after ovulation in the luteal phase of the female cycle [8], has for a long time also been discussed as being an osteo-anabolic hormone inducing the proliferation and the differentiation of osteoblasts [9,10,11,12,13]. In 1999, Katzburg described the differences in bone physiology between males and females and identified the postmenopausal reduction of osteoblasts as one of the contributing factors for the development of osteoporosis in women [14].

In the 1990s, the leading edge of the Baby Boomer Generation in the USA reached middle age. For this reason, the National Institutes of Health planned to undertake extensive studies on women’s health in mid-life. In 2001, the Stages of Reproductive Aging Workshop (STRAW) was held with the aim of developing uniform international scientific standards for the definition and classification of the stages of reproductive aging. During this workshop, the 1996 WHO definitions were re-evaluated, resulting in the following STRAW nomenclature.

Perimenopause is thus defined as the period of time from the onset of menstrual variability (when menstrual cycles still had been regular before) until 12 months after the last spontaneous menstruation. The time period until the final menstrual period (FMP) is also referred to as menopausal transition [15,16]. According to the STRAW staging system, perimenopause (or the menopausal transition) consists of an early and a late stage. The different stages are defined by the degree of variability of menstrual cycle length, with late perimenopause being characterized by ≥2 missed menstrual cycles or ≥60 days of amenorrhea.

In subsequent years, various authors have confirmed that the decline in bone mineral density leading to postmenopausal osteoporosis begins years before menopause. Even though there are discrepancies between investigational conclusions regarding the onset of bone mineral density loss, numerous studies provide evidence for it occurring before the final menstrual period. Longitudinal studies have established a particularly pronounced decline of bone density during perimenopause and have shown that it continues into the early years of post-menopause before subsiding again after four to six years [1,2,3,4,5,17]. During the past decade, quantitative computed tomography (QCT) has gained significance as a complementary volumetric method to the DXA measurements used in these studies for investigating the onset and the specific causes of perimenopausal bone density loss, as only QCT allows for the separate assessment of trabecular bone mass, which rapidly reacts to hormonal stimuli, and cortical bone mass, which is less responsive to hormonal changes [18,19]. Our own recent investigations over a period of nine years revealed a significantly increased decline in trabecular bone density in perimenopausal women, also compared to women in the early stage of post-menopause. Thus, also in the investigated patient population, the greatest loss in bone density was observed already before women entered menopause [20,21,22,23,24]. Our previous study found a decrease in trabecular bone mineral density of −38.8 mg/cm^2^ (−28.9%) over a period of 9 years in women undergoing menopausal transition. 

Perimenopause has—in some of the 20 patterns known for the menopausal transition by now—the unique feature of declining rates of ovulation. Rates of ovulatory cycles decline from 60% 7 years before to 5% 6 months before the final menstrual period. Progesterone therefore may or may not be present, while estrogen supply is still sufficient to cause endometrial proliferation leading to menstruations. For this reason, perimenopause is the ideal life phase to investigate the effects of presence or absence of endogenous progesterone. Since, however, progesterone is only produced in relevant amounts several days after ovulation, in order to differentiate between ovulation and anovulation, it is necessary to pinpoint the timing of serum sampling to 6–9 days after the LH peak or 4–7 days after ovulation. Due to this diagnostic and methodological difficulty, large studies such as SWAN and others only investigated the first week of the menstrual cycle. It is much more demanding to study the effects of variations in the second half of the menstrual cycle, which we shall call “the luteal phase”, despite the fact that a true luteal phase only follows an ovulation, because there is yet no proper term to describe the second half of an anovulatory cycle.

The here presented PeKnO study (*Perimenopausale Knochendichte und Ovulation*; Perimenopausal Bone Density and Ovulation) was designed to systematically investigate the aforementioned factors and their influence on perimenopausal bone density.

## 2. Materials and Methods

The PeKnO study protocol was reviewed and authorized by the ethics committee of the Technical University of Munich (Project nr. 1215/05 I, 14 January 2005).

Via clinic notice boards and newspaper ads, the study recruited healthy women aged ≥45 years with menstrual cycle lengths of ≤42 days who had not taken any exogenous hormones within a period of 6 months prior to baseline (with the exception of thyroid hormones). Subsequent to an extensive assessment of the patients’ medical histories and blood sampling for prolactin, vitamin D, calcium, and TSH levels to exclude bone-relevant subclinical changes, each participant received a commercially available electronic fertility monitor and the appropriate test sticks. Furthermore, participants were instructed in detail on how to use the device every morning for the self-assessment of menstrual cycle lengths, ovulation probability, and the time of probable ovulation. Based on the device data relating to the probable time of ovulation, participants were scheduled for blood sampling in the mid-luteal phase of every 6th cycle (baseline and cycles 6, 12, 18, and 24), as only at this time point ovulatory and anovulatory cycles may be distinguished by means of serum progesterone levels. At the aforementioned five time points of examination, serum hormonal concentrations (LH, FSH, progesterone, 17β-estradiol, and cortisol) were measured together with serum levels of bone turnover markers (osteocalcin, BAP, and CTX). During these visits, an intermediate anamnestic assessment was performed and data from the fertility monitor were transferred to the PC of the working group (see Figure 1: study design).

At baseline and at 24 months, bone mineral density was measured by means of quantitative computed tomography (QCT) at the Department of Diagnostic and Interventional Radiology of the TUM—University hospital Klinikum rechts der Isar in Munich. BMD measurements were acquired using the Somatom Plus 4 computed tomography system (Siemens, Erlangen, Germany) and an in-scan calibration phantom (Osteo calibration phantom; Siemens, Erlangen, Germany). Trabecular bone mineral density (BMD) is measured by means of so-called Pac-Man-shaped regions of interest (ROIs) on CT slices and is expressed in mg of calcium hydroxylapatite (CaHA) per mL of bone. Trabecular bone density values in QCT measurement are classified as follows: values of >120 mg CaHA/mL are defined as normal bone mineral density, values of 80–120 mg CaHA/mL are defined as osteopenia, and values of <80 mg CaHA/mL are defined as osteoporosis.

All levels of hormones and bone turnover markers were determined in serum specimens collected from fasted patients until 11 a.m. at the latest. Blood specimens were drawn in the second half of the menstrual cycle. The time period of 6–9 days after the first appearance of the ovulation symbol on the monitor display was regarded as the optimal time for drawing blood samples. In women where the monitor display did not show any increase in hormonal levels, blood samples were collected between day 19 and 22 of the respective menstrual cycle. Cycles during which blood samples were not drawn within the correct time interval could not be reliably assessed as ovulatory or anovulatory and were therefore excluded from the analysis. As an exception, prolonged cycles of >42 days were included in the analysis and were, by increased probability, considered as anovulatory, even if blood had not been collected within the optimum time interval.

### 2.1. Menstrual Cycle Assessment by Means of Fertility Monitors

Upon recruitment, each participant received a hand-held electronic fertility monitor with the appropriate single-use urine test sticks for documenting all menstrual cycles over the course of the 2-year study participation. On the first day of a new menstrual cycle, the monitor is set to day 1. Thereafter, the respective day of the menstrual cycle is displayed on the monitor. From day 5 onwards, the monitor requests participants to perform urine tests on at least 10 consecutive days. A test stick is dipped in the first morning urine and inserted into the device. The monitor measures the ratio of luteinizing hormone (LH) and estrone-3-glucuronide (E3G) and semi-quantitatively displays a low, high, or maximum hormone concentration. When the LH and E3G ratio is elevated, the monitor shows an ovulation symbol over two consecutive days, signaling that ovulation will probably occur 24–36 h later. The device may store data on 6–8 cycles, thus data could be transferred to the PC of the working group at every visit via chip cards and the appropriate software. Registered monitor data yielded menstrual cycle lengths, the time points of probable ovulation, and thus luteal phase durations, and allowed for distinguishing between ovulatory and anovulatory cycles.

### 2.2. Data Collection and Statistical Analysis

Data were collected in Microsoft Windows Excel. Statistical analyses were performed with the statistical software SPSS version 18.0 and in collaboration with the Institute of Medical Statistics and Epidemiology (IMSE) of the Technical University of Munich. Mean values and standard deviations were calculated for the continuous variables. Differences were analyzed for statistical significance by non-parametric tests: paired samples were tested by Wilcoxon’s signed-rank test and unpaired samples by Mann–Whitney U test. Correlations between variables were analyzed by Spearman’s rho test. The threshold for statistical significance was set at 0.05 and tests were two-tailed.

## 3. Results

Out of 72 women who were recruited, 49 completed the five 2-year follow-up visits and were available for analysis (see CONSORT diagram in Figure 2). Fourteen women dropped out of the study prematurely or were excluded. Reasons included personal reasons (n = 7), onset of continuous amenorrhea immediately after the start of the study (n = 3), initiation of hormonal therapy for marked climacteric disorders (n = 2), a rise in prolactin levels as a side effect from doxepin administration (n = 1), and bilateral adnexectomy (n = 1). For two patients, no delta BMD could be derived, despite five complete examinations in both: for one patient, the first BMD measurement was lost due to technical reasons and the second patient failed to have her second BMD scan.

From each participant, the following data were available: two bone density measurements, one baseline laboratory test, clinical history data from the five study visits during which blood samples were obtained, and the menstrual cycle data recorded by the monitoring devices. Prior to the first bone mineral density measurement, all women had normal serum levels of PRL, calcium, and TSH, and vitamin D concentrations were at ≥10 ng/mL.

Participants were on average 48.1 (±2.4) years old (range: 45–53 years). Upon inclusion in the study, 40 women (81.6%) showed a body mass index (BMI) within the normal range. Nine participants (18.4%) had a BMI of >25 kg/m^2^. None of the women was underweight. After two years, 33 women (67.4%) showed a normal body weight and 16 (32.7%) were overweight. Over the course of the 2-year study period, the average BMI increased from 24.0 kg/m^2^ (±3.5) at baseline to 24.4 kg/m^2^ (±3.7) at the end of the study.

All participants were in perimenopause. Twenty-five women showed a slight variability in menstrual cycle lengths with cycles varying by more than 6 days. There were, however, no menstrual cycles lasting longer than 60 days. These women were considered to be in early menopausal transition. Eighteen participants revealed increasingly irregular and prolonged menstrual cycles (>60 days). They were in late menopausal transition. Five of the forty-nine women reached their final menstrual period during the study period. One participant had consistently regular cycles varying by ≤6 days.

### 3.1. Ovulatory and Anovulatory Cycles

#### Monitored Cycles

By means of the fertility monitoring devices, a total of 1030 analyzable cycles were documented from the 49 study participants. Over the 2-year study period, an average of 21 (±7.48 SD) cycles per participant were registered. The mean cycle length of all registered cycles was 32.3 days (SD ± 22.30). In 596 recorded cycles (57.86%), an ovulation was deemed probable as per monitor display. Monitors identified 434 cycles (42.14%) as anovulatory. Ovulatory cycles were categorized in accordance with the formula proposed by J. Prior, “LPL = CL – Ovmax -2”, into cycles with a normal (10–18 days) and cycles with a shortened (5–9 days) duration of the luteal phase. In 3% of cases, there was no information on luteal phase duration due to unclear monitor display. A total of 471 (79%) of the 596 ovulatory cycles had a luteal phase of normal length, and in 92 (21%) ovulatory cycles, the luteal phase was shortened.

Ovulatory cycles lasted 26.83 (±3.63) days on average (see Figure 3). The 434 anovulatory cycles which had been registered by the monitoring devices were classified according to their length into normal, short, or long. A total of 268 cycles (61.75%) had a normal length of 20–42 days, 47 (10.83%) were shortened with a length of <20 days, and 119 (27.42%) were prolonged. Anovulatory cycles were more frequently longer and varied stronger in length than ovulatory cycles (MV 44.03 (±49.07), shortest cycle 11 days, longest “cycle” 355 days).

Of 245 scheduled blood specimen collections, 228 (93.1%) were carried out. Seventeen blood sample collections were not done due to participants’ personal reasons or due to long phases of amenorrhea. A total of 189 blood sample collections (77.14%) were undertaken during the second half of the cycle and 39 (15.92%) were done less than four days before the beginning of the next menstruation. These were not included in the analyses as no distinction between ovulatory and anovulatory progesterone levels was possible due to the premenstrual drop in hormone concentrations.

According to the 189 valid blood samples collected at the correct time, 111 cycles (58.7%) were ovulatory and 78 (41.3%) were anovulatory. The percentage of ovulatory cycles continually declined from 67.35% to 32.65% over the 2-year course of the study. Blood tests after the second collection during the 6th menstrual cycle revealed that 55% of women had ovulatory cycles with a normal-length luteal phase. One year later, this percentage was at 32.65%, and after two years it was at 24.49%. Eight percent of all cycles were ovulatory with shortened luteal phases (5 to 9 days) and this percentage remained constant over the first four blood draws. At the last blood sample collection, however, no menstrual cycles of this kind were observed at all. The number of cycles where the length of the luteal phase was unknown (anovulatory cycles by monitor display, progesterone > 6 ng/mL) varied between 4% and 22%.

Generalized linear models revealed that the ratio between anovulatory and ovulatory cycles after one year (at the 3rd blood specimen collection) was higher by a factor of 1.4 than at baseline (Odds Ratio: 1.365; 95% CI (0.70; 2.68)). After 2 years (at the final blood sample collection), this factor had increased to almost 4 (Odds Ratio: 3.937; 95% CI (1.88, 8.26)).

Mirroring the decrease in the number of ovulations, the overall percentage of anovulatory cycles rose from initially 28.6% to 51.0%. At the beginning, 16.3% of menstrual cycles were anovulatory by blood test and of normal length (20–42 days). This percentage slightly decreased over the course of the study, however, amounted to 22.45% as per the final blood tests. By contrast, the percentage of anovulatory, prolonged cycles steadily increased from initially 12.24% to finally 28.57%. Overall, more than 50% of all menstrual cycles were anovulatory after 2 years (Figure 4).

### 3.2. Bone Mineral Density Measurements

The first assessment of bone mineral density showed an average trabecular bone mineral density of 139.45 mg CaHA (±24.6 SD). Forty women (81.6%) had a normal bone mineral density (>120 mg CaHA/mL) and 9 participants (18.4%) were osteopenic (80–120 mg CaHA/mL). When bone mineral density was measured for the second time at 24 months, the mean value was at 134.0 mg CaHA/mL (±24.55). Thirty-four participants (68.75%) were still within the normal range, 15 women (31.25%) showed a decreased value. Over the course of 2 years, the average loss of bone mineral density was 6.13 mg CaHA/mL (±8.9) and the percentage of BMD loss was −4.26% (±6.45) (Figure 5).

To compare the endocrinologic characteristics of participants with different rates of changes in bone density, participants were stratified into different BMD subgroups according to Gass et al. [19] and Müller et al. [25], who in their studies had referred to patients whose bone mass decreased by >3.5% per year (=>7% in 2 years) as “fast bone losers”. Group I therefore comprised women whose bone mineral density increased within the 2-year study period (n = 9), group II included participants with a mild decrease in bone mass of ≤7% within the 2 years (n = 26), and group III comprised women with a bone mineral density loss of >7% in 2 years (n = 12). This group of so-called fast bone losers accounted for 25% of the total population. According to Riis and Christiansen, who followed up a cohort of women over a period of 15 years, a low bone mass and a fast rate of bone loss are equally important risk factors for future fractures, each with an odds ratio of 2 [26].

There were no differences between the three groups (I, II, III) regarding participants’ age and BMI.

### 3.3. Ovulation and Changes in Bone Mineral Density

From all evaluable cycle data registered by the monitoring devices and obtained from blood tests, we calculated the percentages of ovulations and compared them with bone density changes. There was a positive correlation between the percentage of ovulatory cycles and the difference in bone density (correlation coefficient: 0.334, *p* = 0.022, Figure 6). Thus, women in whom 80% of cycles had been ovulatory during the observation period experienced no loss in bone density (Figure 6). By contrast, women with an ovulatory rate of only 20% showed a loss in bone density of around 10% within 2 years.

### 3.4. Fast Bone Losers

Twelve of forty-nine participants lost more than 7% of their bone density within the 2-year study period. These women are referred to as “fast bone losers” [19,26,27]. As far as baseline and end-of-study BMI values are concerned, these women did not differ from the rest of the population (BMI_0 of group III: 24.22 kg/m^2^ vs. BMI_0 rest: 24.10 kg/m^2^). Moreover, women in all three groups were comparable regarding their age.

Nine of these twelve women entered the late menopausal transition stage during the observation period. Three women were in early perimenopause. By comparison, women in group I, who showed no decline in BMD, tended to be at the beginning of the menopausal transition.

With an average of 5.23 IU/l, LH levels in women whose BMD had increased (group I) were markedly lower than in women with a BMD decrease. In participants belonging to group II, whose BMD had declined by ≤7% within 2 years, average LH levels were 13.24 IU/l and thus lower than in group III, where the average value was 20.24 IU/l (>7% BMD loss; *p* < 0.001). BMD groups also correlated with FSH, which was lowest in Group I (mean 6.7 IU/l) and highest in group III (mean 33.4 IU/l; (*p* < 0.001). When comparing the median values from the final blood tests (fifth blood sample collection), this difference became all the more apparent: while group I still showed a very low average FSH level (4.8 IU/l), FSH levels in group III had already markedly increased (46.7 IU/l).

We found a negative correlation between BMD groups and 17β-estradiol levels as well as between classification by BMD and progesterone levels. Group I showed the highest average E2 level (222.7 pg/mL), while in the fast bone losers group, the mean E2 level was at 125.2 pg/mL (*p* < 0.001). The difference between the fast bone losers and women with a smaller decline in bone mineral density (group II), was, however, smaller in comparison (125.3 pg/mL vs. 153.2 pg/mL) (Table 1).

Differences for progesterone were more pronounced: while group I showed the highest level with a median of 10.8 ng/mL, progesterone concentrations in group III were lowest at 1.1 ng/mL and clearly within the anovulatory range (*p* = 0.005, Table 1).

There were no between-group differences regarding bone turnover markers and cortisol levels. In the group of women whose BMD had increased, levels of the bone formation marker osteocalcin were slightly higher than in both groups with BMD loss (group I: 17.1 ng/mL vs. group II or group III: 16.9 ng/mL and 16.5 ng/mL). This difference, however, was not statistically significant. In group I, concentrations of bone-specific alkaline phosphatase were on average lower (8.7 μg/mL) than in groups II and III (9.6 μg/mL and 9.2 μg/mL). Again, differences were not statistically significant. There were no differences between groups with regard to the bone resorption marker CTX (C-terminal telopeptide) (group I: 0.26 ng/mL; group II: 0.25 ng/mL; group III: 0.26 ng/mL; correlation coefficient: 0.001; *p* = 0.99). In the three BMD groups, average concentrations of cortisol were also very similar (group I: 16.9 μg/dl; group II: 15.7 μg/dl; group III: 16.5 μg/dl; correlation coefficient: 0.007; *p* = 0.90).

Finally, differences between BMD groups with respect to the relative percentage of ovulatory cycles were investigated. Women of group I, whose BMD had increased during the study period, showed the highest percentage of ovulations (61.27%). In women of group II (bone mineral density loss ≤7% within 2 years), 50.7% of menstrual cycles were ovulatory and participants in the fast bone losers group (group III, bone mineral density loss >7% within 2 years) showed the lowest mean percentage of ovulatory cycles (39.9%). Due to the small size of the subgroups, these differences did not reach statistical significance (*p* = 0.10).

## 4. Discussion

The majority of studies on perimenopausal bone change have been conducted using dual (energy) X-ray absorptiometry (DX) of the hip and spine. DXA provides an areal measurement of mineral content including both cortical and trabecular bone. Trabecular bone, which is more responsive to hormonal changes, provides a more sensitive assessment of bone changes and may result in earlier detection of bone loss (24). The QCT measurements employed in the PeKnO study enable a more differentiated analysis of bone changes than any studies using DXA.

Despite the existence of the STRAW criteria for standardization of perimenopause according to distinct hormonal or menstrual cycle criteria, they have not been consistently used and international standardization for studies has not been achieved. When they were applied, hormonal criteria were mostly limited to the early follicular phase, i.e., the first 7 days of the menstrual cycle.

Only during the luteal phase of the menstrual cycle can ovulatory cycles be distinguished from anovulatory cycles. The present data complement previous work by providing results from the second halves of the menstrual cycles of perimenopausal women and their association with changes in bone mineral density during this, in endocrinological terms, unstable period of life. The presented data imply an influence of ovulatory processes and or progesterone on bone remodeling. This influence has, until now, not been assessed in investigations mostly focusing on the early follicular phase (days 2–7, e.g., the SWAN study). It is known from previous studies that during the 7 years before menopause, the ovulation rate declines from approximately 60% to 7% during the last 6 months of cyclical ovarian activity [27,28]. Even though the levels of endogenous estradiol were only half as high in fast bone losers compared to group I, they had, with an average of 129 pg/mL (median 79 pg/mL), still been within the target range of hormone replacement therapy or even higher until study end. However, median progesterone levels, which had initially been higher by several dimensions, were found to be lower by a factor of 10 in fast bone losers as compared to women with increasing BMD (Table 1). In vitro data in long-term primary osteoblast cultures from perimenopausal women who had undergone hip replacement for arthrosis showed a dose-dependent enhancement of differentiation and increase in alkaline phosphatase by progesterone in 4-week experiments [29]. 

A recent systematic review and meta-analysis of controlled trials with direct randomization in postmenopausal women showed that estrogen-progestin therapy caused a greater increase in spinal bone mineral density than estrogen therapy alone—a systematic review and meta-analysis of controlled trials with direct randomization [30].

## 5. Conclusions

Clinical studies on various progestins and bone mineral density changes have shown inconsistent results. Previous studies which were conducted in perimenopausal women predominantly assessed endocrinological criteria during the first week of the menstrual cycle (days 2–7) and did not distinguish between ovulatory and anovulatory cycles. According to our data, this distinction may account for a yearly trabecular bone loss of up to 5%. This rate exceeds the therapeutic effects seen in some bone treatment trials, including bone density studies in women after premenopausal or perimenopausal breast cancer.

Apart from the age-related physiological decline of ovulatory activity, various other influencing factors are known to also cause the rate of anovulatory cycles to increase. These are high levels of stress, eating disorders, and irregular sleep-wake rhythms. In our research, we have not yet investigated in which way the aforementioned factors influence the ovulation rate in perimenopausal women and whether a potential influence translates into differences in bone resorption rate.

The results of the present work prompt the question as to whether a relative progesterone deficit might be a causative factor for perimenopausal bone loss. Our own research in human osteoblasts implied that progesterone may enhance the differentiation of osteoblasts in a dose-dependent manner [30]. Apart from the indisputable antiresorptive action of estrogen in bone metabolism, the present results suggest that it is possible that the decreased ovulatory activity during perimenopause also minimizes anabolic processes in the bone, an effect which persists into post-menopause.

This study was not able to differentiate whether progesterone itself or other factors associated with ovulatory activity were responsible for the detected differences in bone change. Further studies will need to show whether exogenous progesterone alters biochemical bone markers and/or bone density in situations without ovulation.

Finally, perimenopausal changes in other organ systems have not been studied sufficiently with regard to the implications of ovulatory decline and progesterone deficiency in the presence of estrogen levels within the therapeutic target range. This lack of scientific evidence should also be corrected for topics such as lower urinary tract disorders, breast cancer risk, cardiovascular changes, sleep disorders, and CNS changes.

## Figures and Tables

**Figure 1 diagnostics-12-00305-f001:**
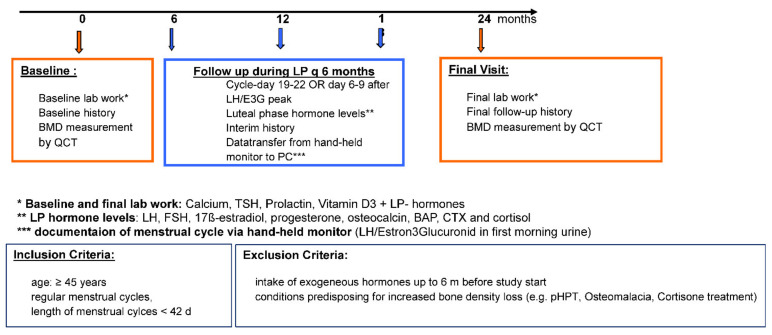
Design of PEKNO-Study.

**Figure 2 diagnostics-12-00305-f002:**
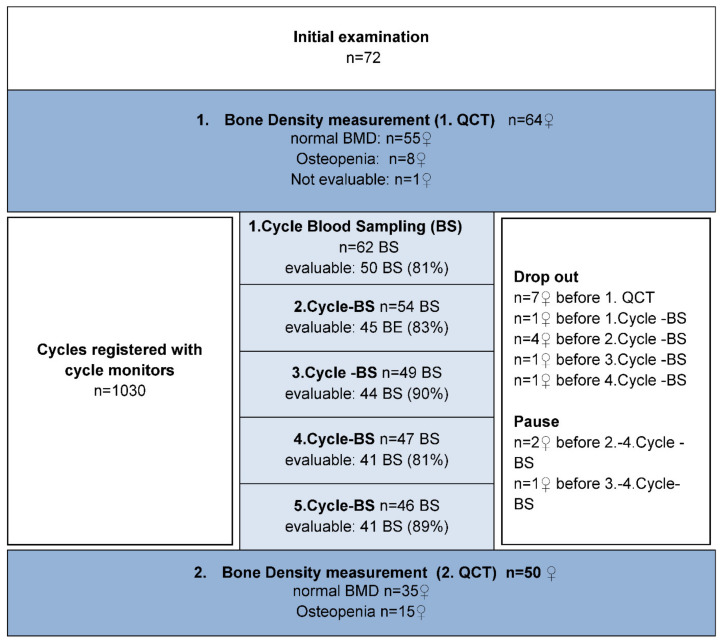
Consort-Diagram of the PEKNO-Study.

**Figure 3 diagnostics-12-00305-f003:**
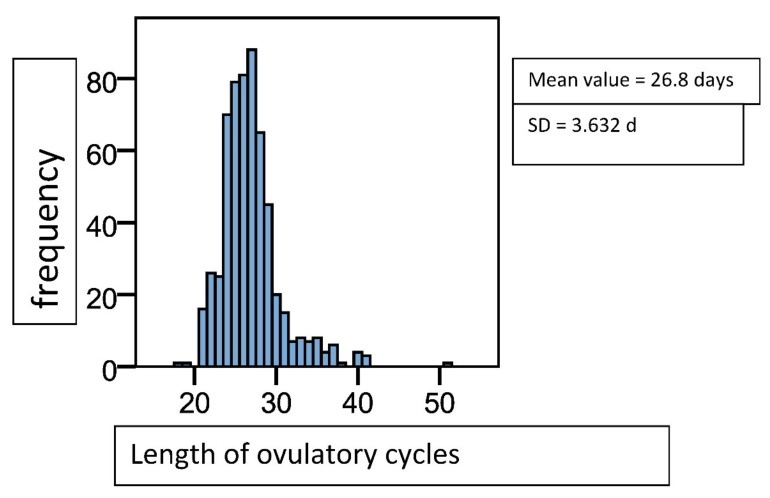
Histogram of cycle length distribution in ovulatory cycles, n = 596. Mean value = 26.8 days, SD = 3.6 d.

**Figure 4 diagnostics-12-00305-f004:**
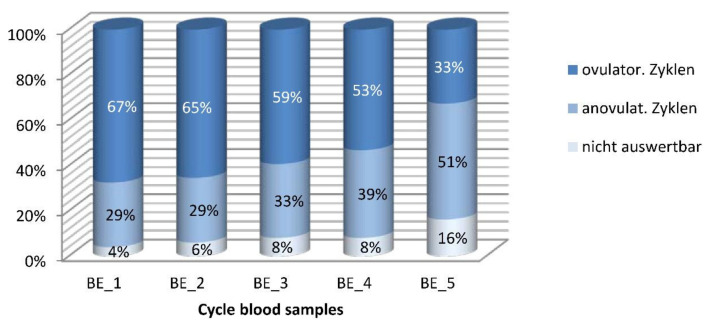
Distribution of ovulatory and anovulatory cycles in percent of the cycles with blood sampling. n = 49♀, n = 221 BS. Cut-off for ovulation was a progesterone value of ≥6 ng/mL. Columns show the distribution of results of the five blood samples, with dark blue top representing ovulatory cycles, light blue middle portion representing anovulatory cycles, and white column-bases showing the proportion of non-evaluable serum samples.

**Figure 5 diagnostics-12-00305-f005:**
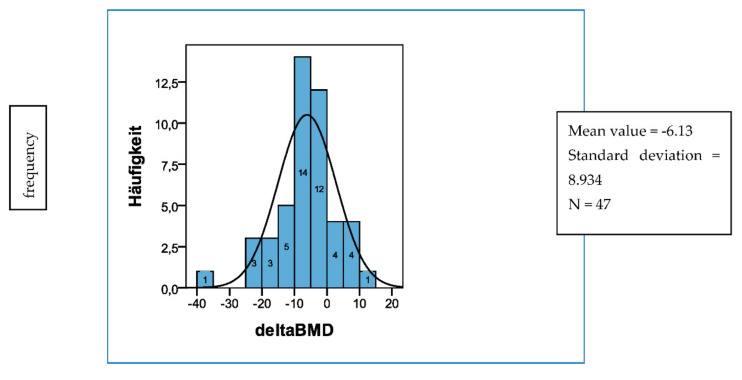
Change of trabecular bone mass density (delta BMD in mg Ca-HA/mL) over 2 years. n = 47; baseline BMD: mean value: 139.45 mg Ca-HA/mL (±24.59); 2-year BMD: mean value: 134.04 mg Ca-HA/mL (±24.55); average reduction of BMD: mean value: −6.13 mg Ca-HA/mL (±8.93) = −4.26%% (±6.45).

**Figure 6 diagnostics-12-00305-f006:**
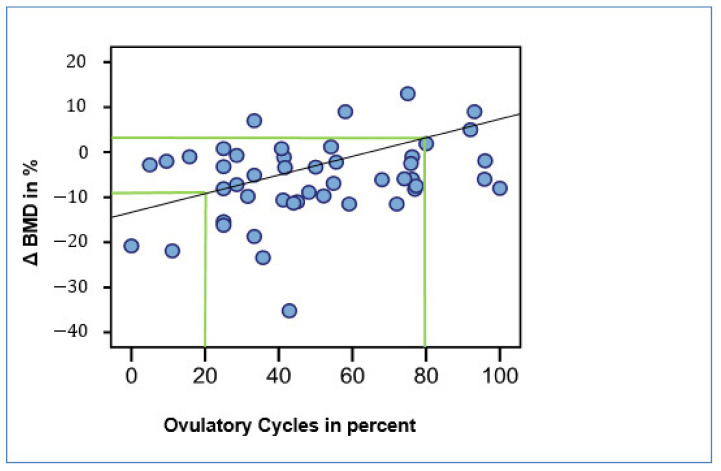
Correlation between ovulatory cycles and change of trabecular bone density (Δ BMD) in percent; n = 47; R: 0.491, r: 0.673, *p* < 0.05.

**Table 1 diagnostics-12-00305-t001:** Luteal phase-hormone values of BMD—Groups I–III. Mean values, standard deviation (±SD) and median of the average of all five blood samples (upper part of the table) of the 2 years of study participation and of the final visit (BS 5) at 24 months only for BMD groups I, II, and III.

	FSH (IU/L)			LH (IU/L)			17ß-Estradiol (pg/mL)			Progesteron (ng/mL)		
	Mean	±SD	Median	Mean	±SD	Median	Mean	±SD	Median	Mean	±SD	Median
All blood samples												
**Group I** *n* = 35 BS	6.68	8.38	4.35	5.23	3.31	4.60	222.73	127.78	191.45	10.50	6.59	10.80
**Group II** *n* = 104 BS	18.17	23.50	7.10	13.24	13.64	6.90	153.15	133.75	132.70	9.40	9.01	8.60
**Group III** *n* = 46 BS	33.37	33.86	13.70	20.24	18.26	10.70	125.27	129.08	84.80	6.74	9.04	1.10
	**FSH** **(IU/L)**			**LH** **(IU/L)**			**17ß-Estradiol** **(pg/mL)**			**Progesteron** **(ng/mL)**		
	**Mean**	**±SD**	**Median**	**Mean**	**±SD**	**Median**	**Mean**	**±SD**	**Median**	**Mean**	**±SD**	**Median**
Last blood sample (BS 5)												
**Group I** *n* = 7 BS	13.51	15.92	4.80	6.15	4.45	4.60	244.39	155.73	190.00	7.34	6.71	4.70
**Group II** *n* = 19 BS	22.49	19.74	20.90	17.62	15.26	11.90	146.28	131.87	101.40	9.28	11.20	7.60
**Group III** *n* = 12 BS	50.60	34.27	46.65	29.19	17.21	28.40	129.13	182.12	78.85	2.9	5.80	0.50

## Data Availability

Data can be found in the archived theses: https://mediatum.ub.tum.de/1081450 and https://mediatum.ub.tum.de/1106225 (accessed on 19 January 2022).

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
