# Peer review of "Perimenopausal Bone Loss Is Associated with Ovulatory Activity—Results of the PeKnO Study (Perimenopausal Bone Density and Ovulation)"

_diagnostics, 2022, doi:10.3390/diagnostics12020305_

Round 1

Reviewer 1 Report

This manuscript deals with a study of bone mass reduction during perimenopause age. The study was based on Perimenopausal Bone Density and Ovulation (PeKnO) of a large number of women monitored for 2 years, and the statistical data were carefully analyzed. In this manuscript, the presented data imply an influence of ovulatory processes on bone remodeling was presented and discussed in detail. This manuscript is potential for publication in Diagnostics after a few minor corrections.

  1. The Introduction can be improved to guide readers on this specific topic.
  2. There are so many abbreviations in this manuscript. It would be better to list up all the abbreviations before the Abstract. This would help readers to find them quickly. For instance, FSH and LH could not be found in the Introduction when they were stated for the first time.
  3. There are several German words in the text and in Figure 4.
  4. Please revise the presentation style of numbers in Figure 3. It should be 26.8 and 3.6 instead of 26,8 and 3,6.

Author Response

Dear Reviewer 1,

thank you for your time and effort to improve our work. We have improved the Introduction to guide readers on this specific topic. Following your suggestions, we have made a list of all the abbreviations, but also introduced them in the Abstract and the Introduction, as you rightly stated it should be. We also adjusted figures 3 and 4 and eliminated the German words (except Hospital names, etc.) in the text.

Reviewer 2 Report

Review of the article Perimenopausal bone loss is associated with ovulatory activity‒ Results of the PeKnO study (Perimenopausal Bone Density and Ovulation)

Congratulations for a useful study with relevant clinical results and for a well-conceptualized and organized article.

§  Please specify the reason for choosing the 7% cut-off for delimiting groups 2 and 3§  Complete in Material and Methods section number and date of ethical comitee aproval§  Please specify the classification used for establish this cut-off of 42 days that is considered by FIGO abnormal uterine bleeding §  Translate in english the fig 4 variables §  Figure 5 is missing§  Rename tables in order, or put back in Results the Table 1a§  The Conclusion section is missing 

The more meticulously designed the study, the more beautiful the introduction and the state of the art it contains, the sharper is the more important part of the article, namely the discussion and conclusion sections.

In my opinion, the results of the study may have practical applicability in many other functional areas affected by the transition to menopause (in this sense please consult the following study: Hormone deficiency and its impact on the lower urinary tract. Berceanu C, Cîrstoiu M, MehedinÈ›u C, Brătilă P, Berceanu S, Vlădăreanu S, BohîlÈ›ea R, Brătilă E. Filodiritto Editore-Proceedings, The 13th National Congress of Urogynecology (UROGYN 2016): 29-38).

Author Response

Response to Reviewer 2:

Dear Reviewer 2, thank you for your compliment and your time and effort to improve our study!

  • The reason for choosing the 7% cut-off was based on the publication by Gass R et al. [19], who found fast losers with a prevalence of 35% in repeated measurements of trabecular bone loss in peripheral QCT of the radius, if a cut-off of > 3,5% loss per year was used. Also, ACR practice guidelines of 2013 reported least significant change (LSC) for QCT of the lumbar spine to be 5%, as opposed to lower rates with DXA-measurements (2-3%). We therefore considered a 7% change in two years to be a good cut-off to discriminate between fast losers and slow losers.
  • The project number and date of the Ethics approval were included in the M&M section.
  • The inclusion criterion cut-off of 42 days was for methodological reasons, since the ovulation-monitors were unable to monitor cycles longer than 42 days. FIGO considers cycles up to 35 days to be usually ovulatory, and labels anovulatory cycles as AUB-O, but we found a substantial number of cycles shorter than 35 days to be anovulatory in this very special perimenopausal cohort. However, this was not the mainstay of the study.
  • Thank you for pointing out the “germanisms” in Figure 4, which have been corrected.
  • Figure 5 was included and the figures renamed in order.
  • The Discussion part of the manuscript has been extended and the Conclusion was added.
